

# Changing trends and emissions of hydrochlorofluorocarbons and their hydrofluorocarbon replacements.

Peter. G. Simmonds[1], Matthew. Rigby[1], Archibold. McCulloch[1], Simon. O'Doherty[1], Dickon. Young[1], Jens. Mühle[2], Paul. B. Krummel[3], L. Paul. Steele[3], Paul. J. Fraser[3], Alistair. J. Manning[4], Ray. F. Weiss[2], Peter. K. Salameh[2], Chris. M. Harth[2], Ray. H.J. Wang[5] and Ronald. G. Prinn[6].

[1] Atmospheric Chemistry Research Group, University of Bristol, Bristol, BS8 1TS, UK
[2] Scripps Institution of Oceanography (SIO), University of California, San Diego, La Jolla, California, USA
[3] CSIRO Oceans and Atmosphere, Aspendale, Victoria 3195, Australia
[4] Met Office Hadley Centre, Exeter, EX1 3PB, UK
[5] School of Earth, and Atmospheric Sciences, Georgia Institute of Technology, Atlanta, Georgia, USA
[6] Center for Global Change Science, Massachusetts Institute of Technology, Cambridge, Massachusetts, USA.

Correspondence to: P.G. Simmonds (petergsimmonds@aol.com)

**Abstract**

High frequency, *in situ* global observations of HCFC-22 ($CHClF_2$), HCFC-141b ($CH_3CCl_2F$), HCFC-142b ($CH_3CClF_2$) and HCFC-124 ($CHClFCF_3$) and their main HFC replacements HFC-134a ($CH_2FCF_3$), HFC-125 ($CHF_2CF_3$), HFC-143a ($CH_3CF_3$), and HFC-32 ($CH_2F_2$) have been used to determine their changing global growth rates and emissions in response to the Montreal Protocol and its recent amendments. The 2007 adjustment to the Montreal Protocol required the accelerated phase-out of HCFCs with global production and consumption capped in 2013, to mitigate their environmental impact as both ozone depleting substances and important greenhouse gases. We find that this change has coincided with a reduction in global emissions of the four HCFCs with aggregated global emissions in 2015 of $444 \pm 75$ Gg/yr, in $CO_2$ equivalent units ($CO_2$ e) $0.75 \pm 0.1$ Gt/yr, compared with $483 \pm 70$ Gg/yr ($0.82 \pm 0.1$ Gt/yr $CO_2$ e) in 2010. (All quoted uncertainties in this paper are 1 sigma). About 80% of the total HCFC atmospheric burden in 2015 is HCFC-22, where global HCFC emissions appear to have been relatively constant in spite of the 2013 cap on global production and consumption. We attribute this to a probable increase in production and consumption of HCFC-22 in Montreal Protocol Article 5 (developing) countries and the continuing release of HCFC-22 from the large banks which dominate HCFC global emissions. Conversely, the four HFCs all show increasing annual growth rates with aggregated global HFCs emissions in 2015 of $329 \pm 70$ Gg/yr ($0.65 \pm 0.12$ Gt/yr $CO_2$ e) compared to 2010 with $240 \pm 50$ Gg/yr ($0.47 \pm 0.08$ Gt/yr $CO_2$ e). As HCFCs are replaced by HFCs we investigate the impact of the shift to refrigerant blends which have lower global warming potentials (GWPs). We also



note that emissions of HFC-125 and HFC-32 appear to have increased more rapidly during the 2011-2015
5-yr period compared to 2006-2010.

**1. Introduction**

Hydrochlorofluorocarbons (HCFCs) were introduced between the 1940s and 1980s as alternatives to
chlorofluorocarbons (CFCs) in some refrigeration and air conditioning applications. Production and
consumption grew rapidly in "non-Article 5" developed countries until the mid-1990s. However, because
they are ozone depleting substances (ODS), HCFCs were included in the 1992 Montreal Protocol
amendment, with a view to eventual phase-out of production and consumption. Subsequently the 2007
adjustments to the Montreal Protocol required an accelerated phase-out of HCFCs in both "non-Article 5"
and "Article 5" developing countries, with a 2013 cap on global production and consumption of HCFCs.
Historically, HCFCs-22, -141b and -142b account for >90% of the total consumption of all HCFCs in
Article 5 countries.
A more rapid phase-out of the HCFCs should result in a faster recovery of the depleted
stratospheric ozone layer, with the additional benefit of mitigating climate change, since these
compounds are also potent greenhouse gases (GHGs). Detailed studies of the rates of atmospheric
accumulation of the HCFCs indicate periods of rapid growth, temporary slowing followed by accelerated
growth (Oram et al., 1995; Simmonds et al., 1998; O'Doherty et al., 2004; Reimann et al., 2004; Derwent
et al., 2007; Montzka et al., 2009; Miller et al., 2010; Saikawa et al., 2012; Fortems-Cheiney et al., 2013;
Rigby et al., 2014; Graziosi et al., 2015). More recently the global growth rates of HCFC-22 and HCFC-
142b have slowed significantly and Montzka et al., (2015) reported that the 2007 adjustments to the
Montreal Protocol had limited HCFC emissions significantly prior to the 2013 cap on global production,
although the atmospheric growth rate of HCFC-141b had almost doubled between 2007 and 2012.
HFCs which have been introduced as replacements for the HCFCs and CFCs have grown rapidly
in abundance since their introduction (Montzka et al., 1996, 2004; Oram et al., 1996; Reimann et al.,
2004; O'Doherty et al., 2009, 2014; Carpenter et al., 2014; Rigby et al., 2014). As discussed by Velders
et al., (2009, 2015) projected HFCs emissions may make a large contribution to future climate forcing if
they are used in the transition away from ODSs.
In this study we focus on high frequency atmospheric measurements (6 -12 per day) of HCFC-22,
HCFC-141b, HCFC-142b and HCFC-124 and their main replacements HFC-134a, HFC-125, HFC-
143a, and HFC-32 from the five core globally-distributed Advanced Global Atmospheric Gases
Experiment (AGAGE) sites with 10-20 year records (Prinn et al., 2000). We have previously estimated



global emissions of HFC-152a ($CH_3CHF_2$) using the same modelling methods discussed in this paper
(Simmonds et al., 2015).

The Ozone Depletion Potential (ODP), Global Warming Potential (GWP) and atmospheric

lifetimes of these eight compounds are listed in Table 1.
Table 1. Lifetimes (yr), Ozone Depletion and Global Warming Potentials (100-yr time horizon) for the
HCFCs and HFCs reported in this study.

| Compound | [a]ODP | [b]GWPs | [c]LIFETIME |
|---|---|---|---|
| **HCFC-22** | 0.055 | 1810 | 12 |
| **HCFC-141b** | 0.11 | 725 | 9.4 |
| **HCFC-142b** | 0.065 | 2310 | 18 |
| **HCFC-124** | 0.02 | 609 | 6 |
| **HFC-134a** | 0* | 1430 | 14 |
| **HFC-143a** | 0* | 4470 | 51 |
| **HFC-125** | 0* | 3500 | 31 |
| **HFC-32** | 0* | 675 | 5.4 |

* see Hurwitz et al., 2015.
Notes: [a] ODPs from the Montreal Protocol, [b] GWPs from (Forster et al., 2007),
[c] Lifetimes, from SPARC, (2013. Report No. 6, WCRP-15/2013).
We combine these observations with a 2-dimensional (12-box) atmospheric chemical transport

model whose circulation is based on observations and on tuning to provide good agreements with global
distributions of reactive and stable trace gases (Cunnold et al., 1983; Rigby et al., 2013; 2014). We then
estimate global emissions which we relate to the global phase-out and adoption schedules of HCFCs and
HFCs, respectively.  We compare these estimated global emissions with HCFC and HFC emission
estimates compiled from national reports to the United Nations Environment Programme, UNEP
(consumption of HCFCs) and United Nations Framework Convention on Climate Change, UNFCCC
(emissions of HFCs), respectively, and Emissions Database for Global Atmospheric Research (EDGAR
v4.2; http://edgar.jrc.ec.europa.eu/, HFCs), using the same techniques reported earlier for these HCFCs
and HFCs (O'Doherty et al., 2009, 2014; Miller et al., 2010; Rigby et al., 2014). We recognise that due to
the historic range of use of these substances (refrigeration, foam blowing, and fire-fighting equipment),
the derivation of emissions from production and consumption data is difficult given the large and long-
lasting banks of these compounds.




We examine the evolution of the changing growth rates of the HCFCs with a view to determining
if the 2013 cap on their production and consumption has been reflected in an accelerated phase-out.
Furthermore, we examine the rapid growth rates of the HFCs and whether these reflect manufacturers of
air-conditioning and refrigeration equipment switching to HFC refrigerant blends with lower GWPs.
HCFC-22 is used in commercial and domestic refrigeration, air conditioning, extruded
polystyrene foams and as a feedstock in the manufacture of fluoropolymers. HCFC-141b and HCFC-
142b are primarily used as foam blowing agents; in addition, HCFC-141b is used as a solvent in
electronics and precision cleaning applications; HCFC-142b is also used as an aerosol propellant and as a
refrigerant.  HCFC-124 has uses in specialized air conditioning equipment, refrigerant mixtures, fire
extinguishers and as a component of sterilant mixtures.
HFC-134a has been used since the early 1990s in vehicle air conditioning systems and other
refrigeration and air conditioning largely to replace CFC-12. Other uses include plastic foam blowing, as
a cleaning solvent and as a propellant. HFCs-125 and -32 have been used as a 50:50 blend (R-410A) in
residential air conditioning systems as well as in 3-component blends with HFC-134a. HFC-125 has also
found application as a fire suppressant agent. HFC-143a is predominantly a component of refrigerant
blends used in commercial refrigeration and in some air conditioning applications.
Although the HFCs are not significant ozone depleting substances (non-zero ODPs; Hurwitz et
al., 2015), as GHGs the HFCs -143a, -125,-134a and -32 have global warming potentials (GWP 100-yr
horizon) of 4470, 3500, 1430, and 675, respectively. The HCFCs, in addition to their ozone depletion
potentials (ODPs), listed in Table 1, are also GHGs with GWPs comparable to the HFCs. This
combination of ozone depletion and climate forcing has provided the impetus for the accelerated phase-
out of the HCFCs.

**2. Materials and Methods**



2.1. AGAGE in situ measurements.
The data used here are compiled from *in situ* measurements at the core AGAGE sites, listed in
Table 2, which shows the time frame when the measurements of individual HCFCs and HFCs began at
each AGAGE site.

2.2. AGAGE Instrumentation and Measurement Techniques
Two similar measurement technologies have been used at AGAGE stations over time, both based
on gas chromatography coupled with mass spectrometry (GC-MS) and cryogenic sample pre-
concentration techniques. The earlier instrument, referred to as the GC-MS-ADS, incorporated an



Adsorption-Desorption-System (ADS) based on a Peltier-cooled microtrap maintained at -50°C during
the adsorption phase (Simmonds et al. 1995; Prinn et al., 2000) and was used for several years at the
Mace Head and Cape Grim sites. These were replaced by another GC-MS instrument, the GC-MS-
Medusa with doubled sampling frequency and enhanced cooling to ~ -180°C, which uses the milder trap
adsorbent HayeSep D, to extend compound selection, and improve measurement precisions (Miller et al.,
2008; Arnold et al., 2012).

The GC-MS-Medusa system is currently deployed at all AGAGE sites used in this study (Table

2). Typically for each measurement the analytes from two litres of air are collected on the sample traps
and desorbed onto a single main capillary chromatography column (CP-PoraBOND Q, 0.32 mm ID × 25
m, 5 µm, Agilent Varian Chrompack, batch-made for AGAGE applications) purged with helium (grade
6.0) that is further purified using a heated getter purifier (He-purifier HP2, VICI, USA). The separation
and detection of the compounds are achieved using Agilent Technology GCs (model 6890N) and
quadrupole mass spectrometers in selected ion mode (initially model 5973, progressively converted to
5975C over the later years). For the field GC-MS-Medusa instruments, ambient air samples are analysed
every 2 hours (c.f. 4 hours for the GC-MS-ADS) and are bracketed by measurements of quaternary
standards to detect and correct for short-term drift in instrument response. The quaternary standards are
whole-air pressurized into 34 L internally electropolished stainless steel canisters (Essex Industries,
USA). They are filled by the groups who are in charge of the respective AGAGE stations using modified
oil-free diving compressors (SA-3 and SA-6, RIX Industries, USA) to ~60 bar (older tanks to ~30 bar).
Cape Grim is an exception, where the canisters used for quaternary standard purposes are typically filled
cryogenically. The on-site quaternary standards are compared weekly to tertiary standards from the
central calibration facility at the Scripps Institution of Oceanography (SIO) in order to propagate the
primary calibration scales and to characterize any potential long-term drift of the measured compounds in
the quaternary standards. Importantly, all of the stations report HCFC and HFC measurements relative to
the SIO (SIO-05, SIO-07 and SIO-14) and University of Bristol (UB-98) calibration scales.

The GC-MS-Medusa measurement precisions for the four HCFCs and four HFCs are estimated

based on the repeated measurements of the quaternary standards. They are typically for HCFC-22 (0.5
ppt, 0.2%), -141b (0.05 ppt, 0.3%), -142b (0.05 ppt, 0.25%) and -124 (0.02 ppt, 1.6%) and for HFC-134a
(0.2 ppt, 0.2%), -125(0.05 ppt, 0.3%), -143a (0.1 ppt, 0.4%) and -32 (0.1 ppt, 0.8%).







Table 2.  Overview of the core AGAGE sites used in this study, their coordinates and periods for which
data are available.

| Site | Latitude | Longitude | ADS Data* | Medusa Data** |
|------|----------|-----------|-----------|---------------|
| **Mace Head, Ireland** | 53.3° N | 9.9° W | 1994-2004 | June 2003-present |
| **Trinidad Head, California, USA** | 41.0° N | 124.1° W | | March 2005-present |
| **Ragged Point, Barbados** | 13.2° N | 59.4° W | | May 2005-present |
| **Cape Matatula, American Samoa** | 14.2° S | 170.6° W | | May 2006-present |
| **Cape Grim, Tasmania, Australia** | 40.7° S | 144.7° E | 1998-2004 | Jan 2004-present |

*   Period of HCFC and HFC data records using GC-MS-ADS.
** Period of HCFC and HFC data records using GC-MS-Medusa.

2.3. Calibration Scales

The estimated accuracy of the calibration scale for the various HCFCs and HFCs is reported

below and a more detailed discussion of the measurement technique and calibration procedure have been
reported elsewhere (Miller et al., 2008; O'Doherty et al., 2009; Mühle et al., 2010). The AGAGE
measurements for the HCFCs-22, -141b, -142b, and HFC-134a, are reported relative to the Scripps
Institution of Oceanography (SIO-05) calibration scale (as dry gas mole fractions in pmol mol$^{-1}$). This
scale is defined through the gravimetric preparation of 13 synthetic primary standards at near-ambient
mole fractions (Prinn et al., 2000) at SIO in 2005. HCFC-124 is derived from the UB-98 calibration scale.
HFCs -143a, -32 are reported relative to the SIO-07 and HFC-125 to the SIO-14 calibration scales.

The overall accuracies of these primary standards sets are liberally estimated at 2% for HCFC-22,

-141b, and -142b, 10%, for HCFC-124 and 1.5% for HFC-134a, and 3% for HFC-125, -143a, and
-32 with the largest fractional uncertainty contributed from the impurities in the starting reagents.
2.4. Selection of baseline data

Baseline *in situ* monthly mean HCFC and HFC mole fractions were calculated by excluding

values enhanced by local and regional pollution influences, as identified by the iterative AGAGE
pollution identification algorithm, (for details see Appendix in O'Doherty et al., 2001). Briefly, baseline
measurements are assumed to have Gaussian distributions around the local baseline value, and an
iterative process is used to filter out the points that do not conform to this distribution. A second-order
polynomial is fitted to the subset of daily minima in any 121-day period to provide a first estimate of the
baseline and seasonal cycle. After subtracting this polynomial from all the observations a standard
deviation and median are calculated for the residual values over the 121-day period. Values exceeding
three standard deviations above the baseline are thus identified as non-baseline (polluted) and removed





from further consideration. The process is repeated iteratively to identify and remove additional non-
baseline values until the new and previous calculated median values agree within 0.1%.

**3.  Modelling studies**

There are several sources of information on production and emissions of HCFCs and HFCs; none of

which, on their own, provides a complete database of global emissions. The more geographically
comprehensive source of information for HFC emissions is provided by the parties to the UNFCCC, but
only includes Annex 1 countries (developed countries). The 2014 database covers years 1990 to 2012 and
emissions are reported in Table 2(II) s1 in the Common Reporting Format (CRF) available at
http://unfccc.int/national-reports/annex ighg inventories/national inventories submissions/items/8108.php.
An alternative inventory estimate was also obtained from the Emissions Database for Global
Atmospheric Research (EDGAR v4.2; http://edgar.jrc.ec.europa.eu/), a database that estimates global
emission inventories of anthropogenic GHGs, including HFCs on a country, regional and gridded basis
up to 2008.

Similar emission estimates are not available for HCFCs, but using HCFC consumption data

published by the Montreal Protocol Secretariat of the United Nations Environment Programme (UNEP,
2016a) we calculate HCFC emissions as described in the supplementary material (1).

Such bottom-up emission estimates of HFCs and HCFCs are based on industry production,

imports, distribution and usage data for these compounds, reported to national governments and thence to
UNEP and UNFCCC. We discuss these independent emission estimates because they are helpful as *a*
*priori* data constraints on our model analysis and to compare them with our observation-based top-down
estimates.

3.1. Global emissions estimates using the AGAGE two-dimensional 12-box model.

To estimate global-average mole fractions and derive growth rates, a two-dimensional model of

atmospheric chemistry and transport was employed. The AGAGE 12-box model simulates trace gas
transport in four equal mass latitudinal sections (divisions at 30-90°N, 0-30°N, 30-0°S and 90-30°S) and
at three heights (vertical divisions at 200, 500 and 1000 hPa). The model was originally developed by
Cunnold et al. (1983) (nine-box version), with subsequent improvements by Cunnold et al. (1994) and
Rigby et al. (2014). Emissions were estimated between 1998 and 2015 using a Bayesian method in which
an *a priori* constraint (EDGAR v4.2) on the emissions growth rate was adjusted using the baseline-
filtered AGAGE observations (Rigby et al., 2011, 2014). Global emissions were derived that included





estimates of the uncertainties due to the observations, the prior and the current best-estimate lifetimes of
these compounds from SPARC (2013); as detailed in the supplementary material in Rigby et al. (2014).

**4.     Results and Discussion**
4.1. Atmospheric Mole Fractions
Based on the output from the 12-box model, into which AGAGE observations had been
assimilated, Figure 1 illustrates the global mean mole fractions for the four HCFCs and the four HFCs,
(the model output was used for "gap-filling" purposes). Figure 2 shows the average annual growth rates.
Global mean mole fractions of HCFC-22, -141b, and -142b have increased throughout the observation
period reaching 234, 24.3 and 22.4 pmol mol$^{-1}$, respectively in 2015. HCFC-124 reached a maximum
global mean mole fraction of 1.48 pmol mol$^{-1}$ in 2007 and has since declined by 23% to 1.14 pmol mol$^{-1}$
in 2015.
The HFCs all show increasing global mean mole fractions and growth rates over the entire period
of observations. In 2015 the global mean mole fractions (pmol mol$^{-1}$) in descending order of abundance
are HFC-134a (83.3), HFC-125 (18.4), HFC-143a (17.7) and HFC-32 (10.5) with growth rates (pmol
mol$^{-1}$ yr$^{-1}$) for HFC-134a (5.6), HFC-125 (2.3), HFC-143a (1.5) and HFC-32 (1.6).



Figure 1. Global mean mole fractions for the four HCFCs and the four HFCs, (the model output was used
for "gap-filling" purposes). Shading in the figure reflects the uncertainty on the mole fractions derived in
the inversion and includes a contribution from random and scale-related measurement errors and
modelling uncertainties (further details are provided in Rigby et al., 2014). Note that HCFC-124, HFC-
143a and HFC-32 use only GC-MS-Medusa data for these calculations; all others use combined GC-MS-
ADS and GC-MS-Medusa data. HFCs shown with individual annual mole fractions.





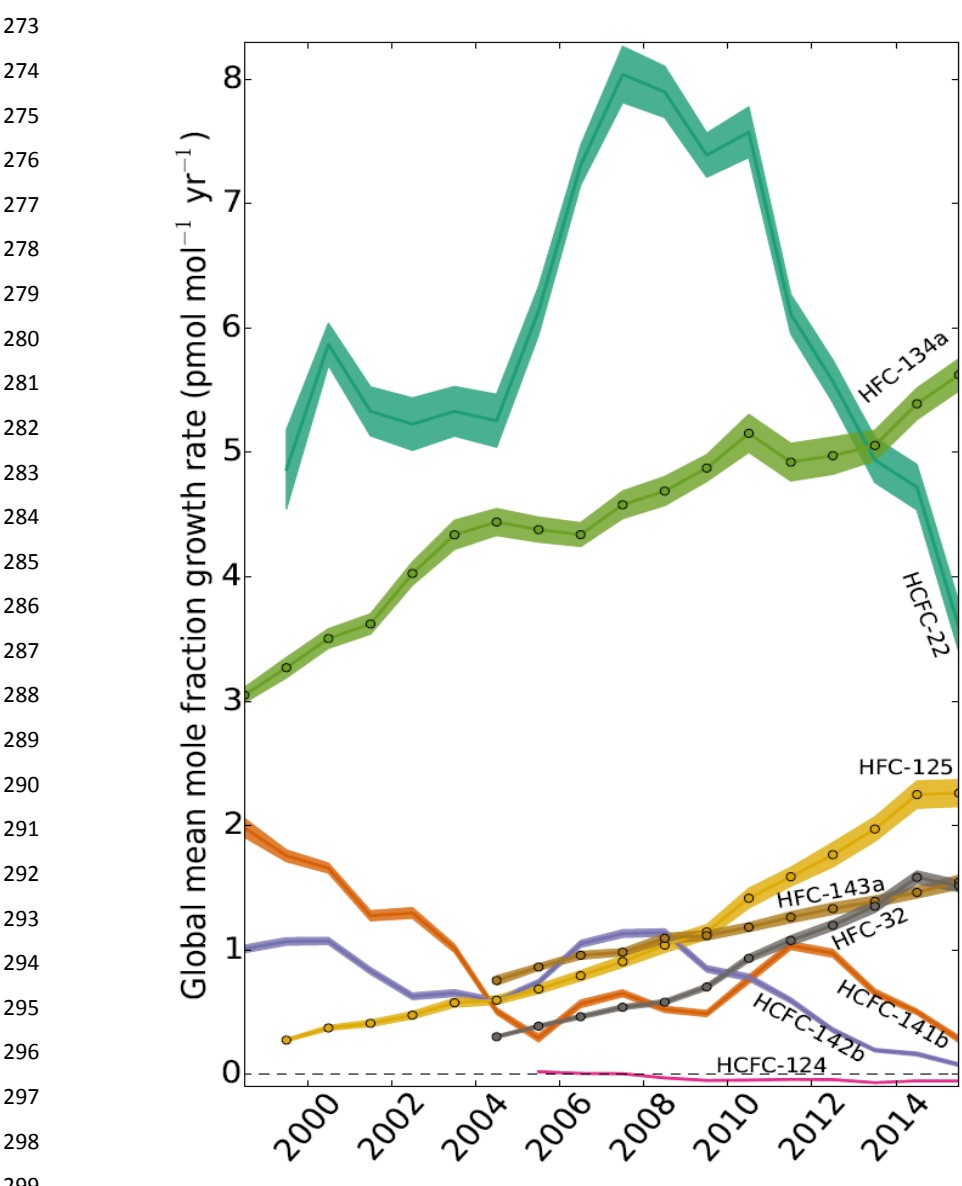

Figure 2. Annual average global mean mole fraction growth rates (pmol mol$^{-1}$ yr$^{-1}$) determined from the 12-box model for the HCFCs and HFCs. Note that HCFC-124, HFC-143a and HFC-32 use only Medusa data for these calculations; all others use combined GC-MS-ADS and GC-MS-Medusa data. HFCs shown with individual annual growth rates.





Global mean HCFC-22 reached a maximum annual rate of increase of 8.2 pmol mol$^{-1}$ yr$^{-1}$ in 2007
and slowed by 54% to 3.7 pmol mol$^{-1}$ yr$^{-1}$ in 2015. The global mean annual growth rates of HCFC-141b
reached a first maximum of 1.9 pmol mol$^{-1}$ yr$^{-1}$ in 1998, followed by a second maximum in 2011 of 1.0
pmol mol$^{-1}$ yr$^{-1}$ and then slowed to ~0.3 pmol mol$^{-1}$ yr$^{-1}$ in 2015, a 70% decline. Similarly, HCFC-142b
reached a maximum in 2008 of 1.1 pmol mol$^{-1}$ yr$^{-1}$, followed by a steep 90% decline to just 0.11 pmol
mol$^{-1}$ yr$^{-1}$. These observations reflect substantial changes in the quantity of HCFCs emitted to the
atmosphere over time with the rates of increase in 2015 considerably slower than their historical highs
and broadly in response to the MP and its amendments.

4.2. Top-down Emission Estimates
4.2.1. Global estimates of HCFCs
Estimated annual global emissions (Gg/yr) of the HCFCs using the 12-box model (emissions
listed in supplementary material 1) and those calculated from consumption reported by UNEP (2016a, see
supplementary material 2) and EDGAR emission inventories are shown in Figure 3 a-d. The blue solid
line represents our model-derived emissions, with the uncertainties shown by the shaded areas.

4.2.1.1. HCFC-22
Model derived global emissions of HCFC-22 increased from 234 ± 35 Gg/yr in 1995 to a
maximum of 383 ± 54 Gg/yr in 2010 increasing by ~10 Gg/yr. Since 2010 global HCFC-22 emissions
have declined by perhaps 6.6% to 357 ± 58 Gg/yr in 2015. Figure 3a includes estimated emissions from
UNEP and other reported global HCFC-22 emissions estimates (Saikawa et al., 2012; Xiang et al., 2014;
Montzka et al., 2015) which all agree within the uncertainties of our estimates. Fortems-Cheiney et al.,
(2013) using observations from multiple networks, an inversion and a new gridded bottom-up inventory
estimated global emissions of 387 ± 9 Gg/yr in 2010, very close to the estimated HCFC-22 emissions
derived in this study.
4.2.1.2. HCFC-141b and HCFC-142b
These two HCFCs have exhibited similar but fluctuating emissions with maxima in 2000 of 63 ±
6 Gg/yr (HCFC-141b) and 31 ± 6 Gg/yr (HCFC-142b) followed by a decline during 2004-2005 to 46 ± 7
Gg/yr (HCFC-141b) and 29 ± 5 Gg/yr (HCFC-142b). HCFC-141b and HCFC-142b emissions then grew
rapidly to new maxima of 68 ± 8 Gg/yr (2012) and 39 ± 5 Gg/yr (2008), respectively. These trends were
again reversed with subsequent declines to 59 ± 10 Gg/yr (HCFC-141b) and 25 ± 6 Gg/yr (HCFC-142b)
in 2015. Montzka et al., (2015), using an independent sampling network, also provided emissions
estimates for HCFCs-141b and -142b in 2012 which are included in Figures 3 b,c and agree within the





uncertainties of our estimates. Inventory emissions estimates reported from EDGAR v4.2 are
considerably lower than all other reported emissions post-2005. Global emissions of HCFC-141b and
HCFC-142b have declined by 2.0% and 30%, respectively, from 2010 to 2015.

4.2.1.3. HCFC-124

Global emissions of this less abundant HCFC had a maximum in 2005 of $6.1 \pm 1.8$ Gg/yr,

followed by a steady decline to $3.0 \pm 0.9$ Gg/yr in 2015, a decrease in emissions over this 12-year time
frame of 51%. There are no bottom-up estimates of HCFC-124 emissions or top-down global estimates to
compare with our results. However, we note that HCFC-124 emissions make only a minor contribution to
climate change in $CO_2$ equivalent units ($CO_2$ e) of 0.002 Gt/yr in 2015.

The combined model derived aggregated global emissions of these four HCFCs in 2015 were 444

$\pm 75$ Gg/yr ($0.75 \pm 0.1$ Gt/yr $CO_2$ e). We estimate that these four HCFCs contribute $55 \pm 1$ mW/m$^2$ to
climate forcing in 2015 with HCFC-22 accounting for 79% of this forcing.


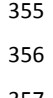
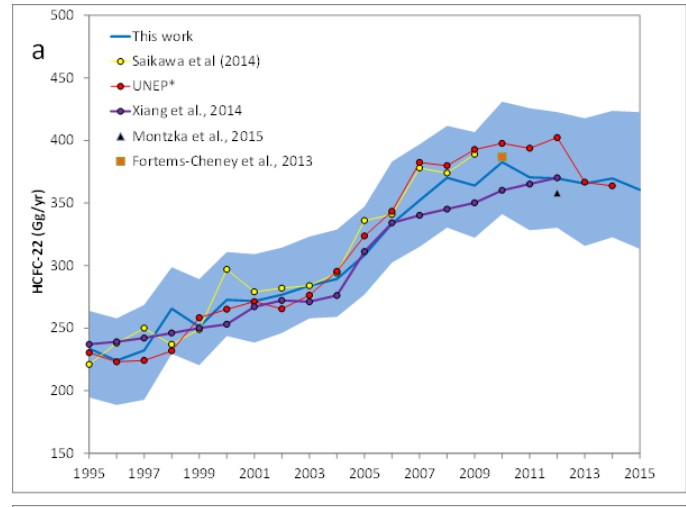


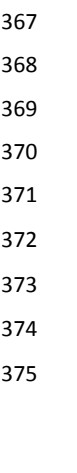
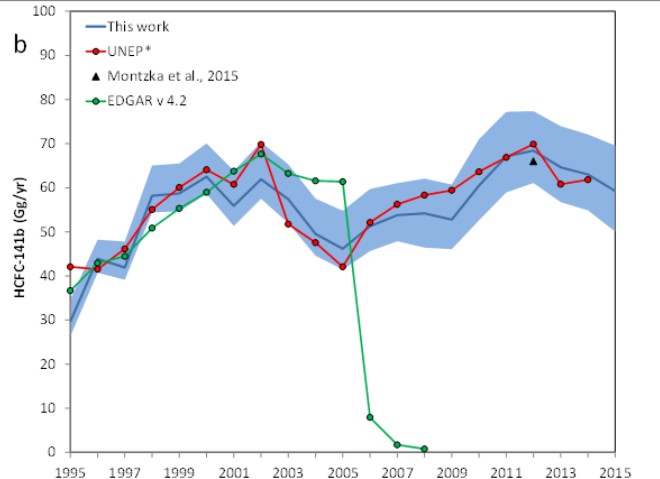





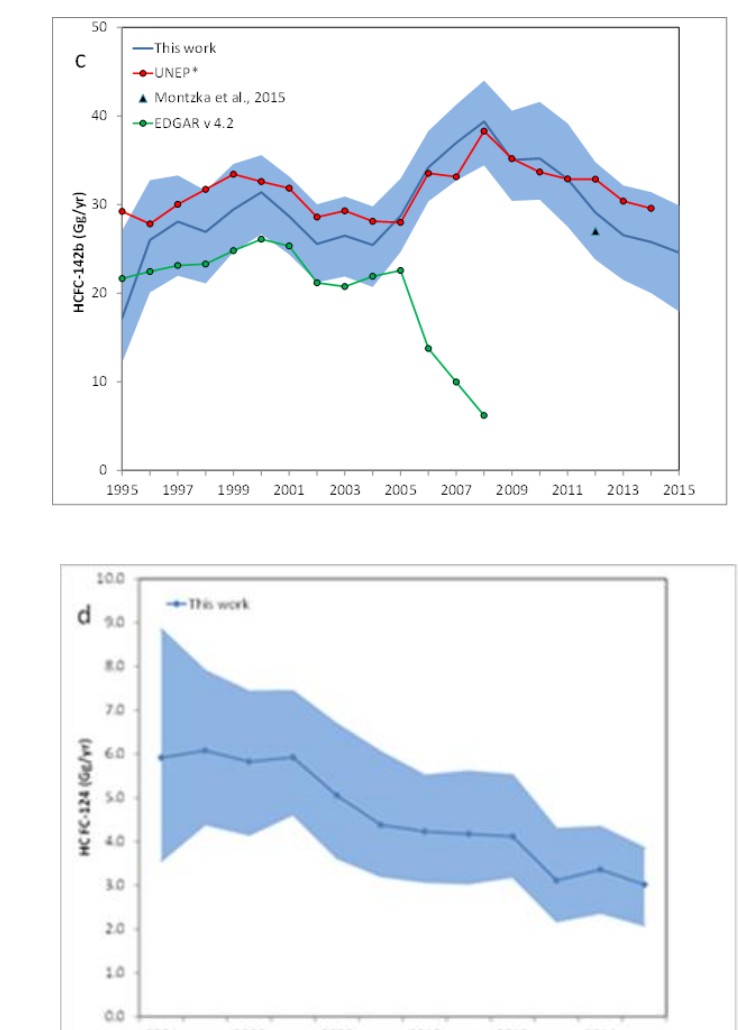

Figure 3 (a-d). Global HCFCs emissions (Gg/yr, blue line) from the 12-box model (solid blue line);
shading, representing 1 sigma uncertainties; emissions derived from UNEP* consumption data (- see
supplementary material 2). EDGAR v4.2 and other works are also shown. Note that HCFC-124 uses only
GC-MS-Medusa data for these calculations; all others use combined GC-MS-ADS and GC-MS-Medusa
data.







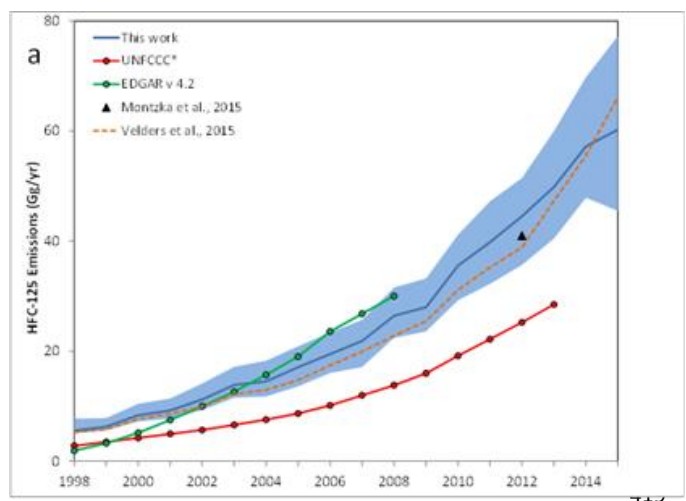

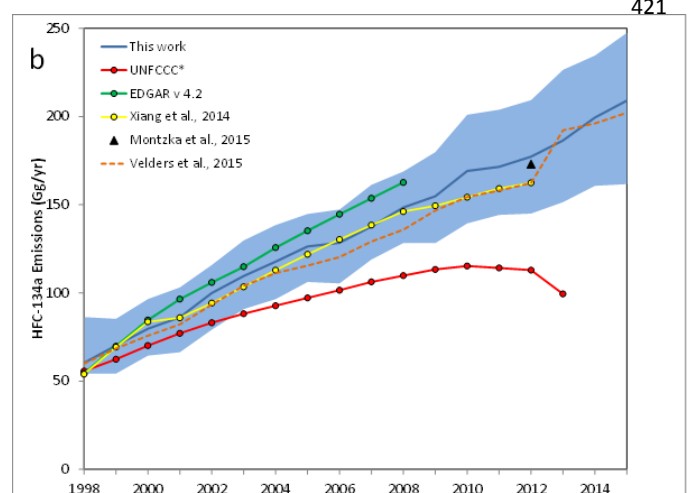














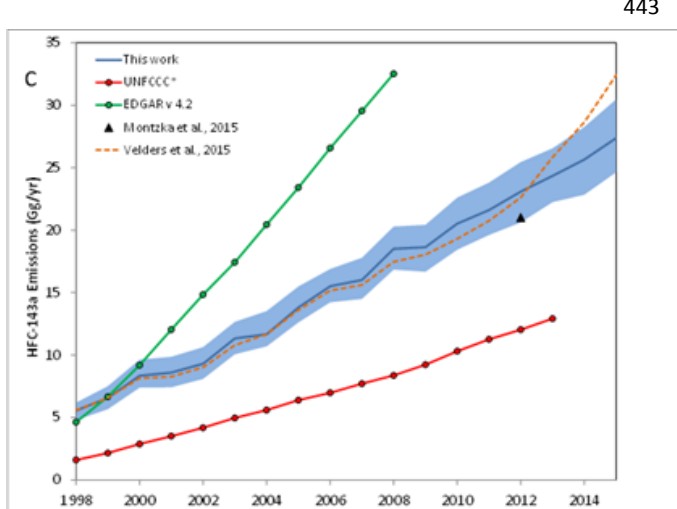

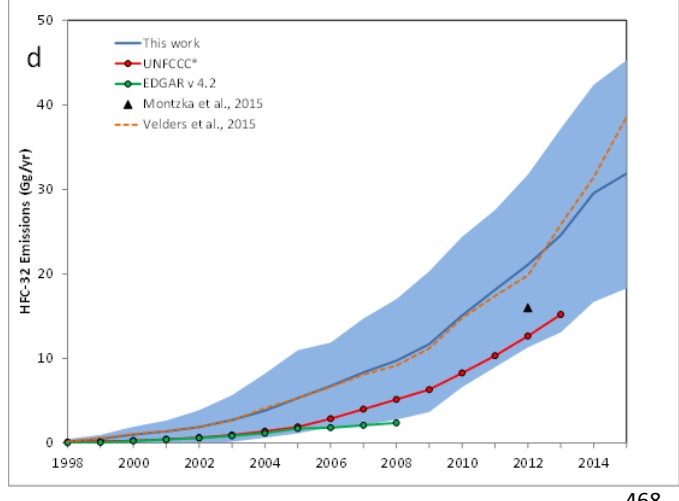



Figure 4 (a-d). Global HFCs emissions (Gg/yr, blue line) from the 12-box model; shading represents 1
sigma uncertainties; UNFCCC* values are the global aggregate of national data reported in UNFCCC
(2016). Note that HFC-143a and HFC-32 use only Medusa data for these calculations; all others use
combined GC-MS-ADS and GC-MS-Medusa data.





4.2.2. Global estimates of HFCs
In contrast to the HCFCs, estimates of HFCs emissions shown in Figure 4 (a-d) (see
supplementary material 1 for actual values) have increased annually over the entire observational record,
reaching maxima in 2015 of $60 \pm 10$ Gg/yr (HFC-125), $209 \pm 43$ Gg/yr (HFC-134a), $27 \pm 3.0$ Gg/yr
(HFC-143a) and $32 \pm 14$ Gg/yr (HFC-32). UNFCCC emission estimates are consistently lower than our
estimates, even for HFC-32 which barely agrees within our uncertainties. EDGAR v4.2 inventory
emissions of HFC-143a post- 2000 are substantially larger than our estimates, but the other three HFCs
are in reasonable agreement (within the uncertainties of our estimates). Recently published HFC
emissions estimates by Velders et al., (2015) are in close agreement with the results from this work and
within the uncertainties of our estimates, except for HFC-143a after 2014.
The combined model derived aggregated emissions of these four HFCs in 2015 were
$329 \pm 70$ Gg/yr ($0.65 \pm 0.14$ Gt/yr $CO_2$ e). We estimate that these four HFCs contribute $21.0 \pm 0.5$
mW/m$^2$ to radiative forcing in 2015, less than half the combined forcing of the 4 HCFCs treated in this
study.

4.3. Overall emissions trends

In Figure 5 we plot individual HCFC and HFC in terms of $CO_2$-e emissions, noting that HCFC-22
is the largest contributor to these emissions and they have declined relatively slowly since 2010. HCFC-
141b and HCFC-142b exhibit declines in $CO_2$-e emissions after 2010, in spite of substantial changes in
emissions over time and dramatic declines in their global mean mole fraction growth rates (see Figure 2).
All of the HFCs have increasing $CO_2$-e emissions over time with HFC-125 and HFC-32 showing the
most rapid increases in $CO_2$-e emissions.




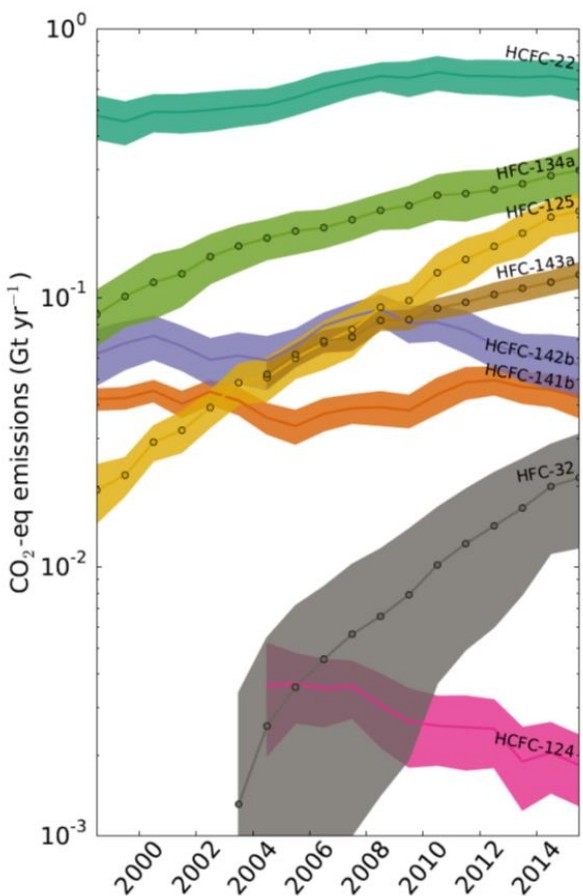



Figure 5. Individual HCFC and HFC carbon dioxide equivalent ($CO_2$-e) emissions (Gg/yr) derived from

the 12-box model. Shading, representing 1 sigma uncertainties. Note that HCFC-124, HFC-143a and

HFC-32 use only GC-MS-Medusa data for these calculations; all others use combined GC-MS-ADS and

GC-MS-Medusa data.

Figure 6 shows the trends in CO2-e emissions when the HCFCs and HFCs are aggregated

together from 2005 to 2015. The figure indicates an increase in aggregate HCFC emissions until around

2010, and subsequent reduction in emissions through 2015. HFC emissions were observed to increase

throughout this period. Using 2010 as a "reference" year, we aimed to compare the relative trends in

HCFC and HFC emissions and determine broadly whether a decline in HCFC emissions was being

matched by acceleration in HFC usage. In order to determine a potential "business as usual" trajectory for

HCFC and HFC emissions post-2010, we first assumed that emissions would continue to follow the 2005





– 2010 growth rates (dashed line). We also examined the potential change in emissions according to the
projections of Velders et al., (2009) and WMO (2010), Chapter 1. S.A. Montzka and S. Reimann. Each of
these projections suggests a growth in HCFC emissions during this period, whereas our observation-
derived estimates show a decline. During this 5-year period, the accumulated difference between the top-
down and projected emissions is -0.67, -0.71 and -0.66 Gt/yr CO2-e (with a 1-sigma uncertainty 0.25 Gt
for each) for the linear projection, Velders et al., (2009) and WMO (2010), Chapter 1. S.A. Montzka and
S. Reimann, respectively. For the HFCs, the linear projection shows more modest growth than we derive,
with an accumulated difference of 0.12 ± 0.15 Gt/yr CO2-e. However, the Velders et al., (2009)
projection exhibits more rapid growth, leading to a difference of -0.27 ± 0.15. When considered together,
we find that the growth in HFC and HCFC emissions has been slower than these post-2010 projections by
an accumulated total of -0.55 ± 0.29 Gt/yr CO2-e and -0.97 ± 0.29 Gt/yr CO2-e for the linear projection
and Velders et al., (2009), respectively.





















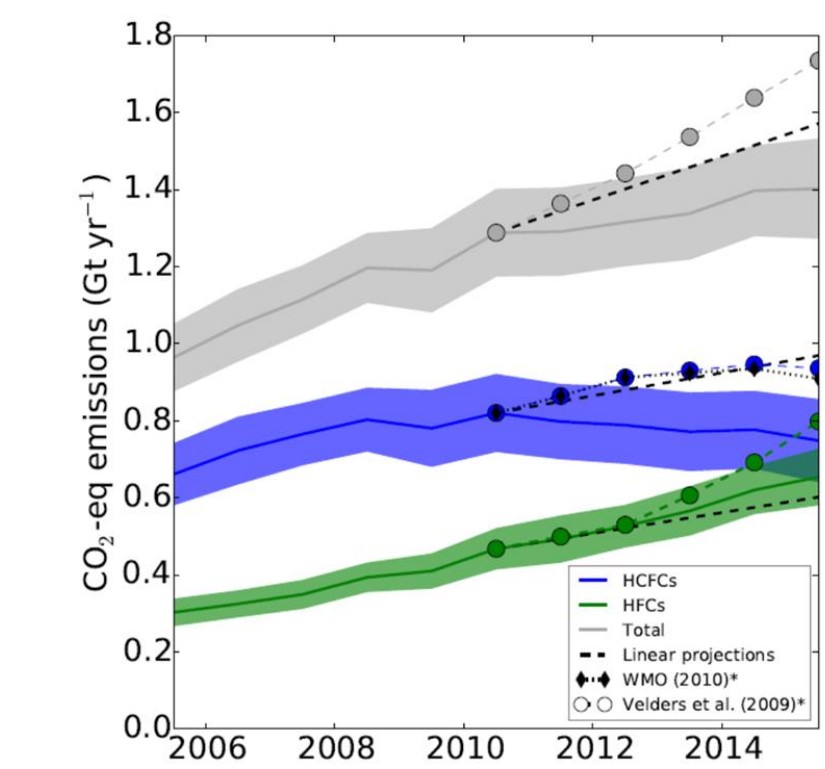





Figure 6. Aggregated HCFCs and HFCs emissions as $CO_2$-e (Gt yr$^{-1}$) solid blue and green lines, respectively; shading representing 1 sigma uncertainties. The grey line and shading represents the total HCFC and HFC $CO_2$-e emissions. The dashed line is a linear projection from 2010 to 2015, based on the 2005 – 2010 emissions growth rate. The dashed lines with the circular data points are the Velders et al. (2009) projection, rescaled to the 2010 emissions rate, and the dashed line with the diamonds show the same but for HCFC emissions projections WMO (2010), Chapter 1. S.A. Montzka and S. Reimann.

In Table 3 we compare the cumulative emissions over two 5-years periods, 2006-2010 when HCFCs were still increasing (with the exception of HCFC-124) and 2011-2015 when all four HCFCs show declining emissions. From the percentage change in emissions between the two periods we note that HCFC-141b emissions have increased by 18% and HCFC-22 by 1.7%. In comparison HCFC-142b and -124 emissions have decreased by 23% and 30%, respectively. Aggregating the four HCFCs we observe a 1.4% increase in the combined emissions between the two 5-yr periods. However, over both 5-yr periods there is an equal contribution of 3.9 Gt $CO_2$-e.

Conversely, global emissions of the HFCs have grown continuously throughout the period of observations with substantial increases between the two 5-yr periods. The largest increases were observed for HFC-32 (143%) and HFC-125 (92%) with smaller increases for HFC-143a (37%) and HFC-134a (28%). In terms of the aggregated HFC emissions we see a 43% increase representing a rise from 1.95 to 2.87 Gt $CO_2$-e, or an additional 0.92 Gt $CO_2$-e between the two periods.

It is also apparent that emissions of HCFC-22 represent 79.3% of the global cumulative HCFC burden during 2011-2015 and HCFC-22 alone contributes 0.45 Gt $CO_2$-e (13.5%) more than all HFCs cumulative emissions during 2011-2015.





Table 3. Global emissions of HCFCs and HFCs in 2015 and a comparison of the cumulative emissions
and the percentage change in emissions over two 5-year periods (2006-2010 and 2011-2015).

| HCFCs | 2015 Emissions (Gg) | 2006-2010 HCFCs Cumulative Emissions (Gg) and $CO_2$-e Emissions (Gt $CO_2$-e) | 2011-2015 HCFCs Cumulative Emissions (Gg) and $CO_2$-e Emissions (Gt $CO_2$-e) | % Change in cumulative emissions between 2006-2010 and 2011-2015 |
|---|---|---|---|---|
| **22** | $357.4 \pm 58.1$ | 1802.0 (3.26) | 1833.2 (3.32) | 1.7 |
| **141b** | $59.3 \pm 9.8$ | 272.5 (0.20) | 322.4 (0.23) | 18.3 |
| **142b** | $24.6 \pm 6.3$ | 180.8 (0.42) | 138.9 (0.32) | -23.2 |
| **124** | $3.0 \pm 0.89$ | 25.4 (0.02) | 17.8 (0.01) | -29.9 |
| Total | | **2281 (3.9)** | **2312 (3.9)** | **1.4** |
| | | | | |
| **HFCs** | **2015 Emissions (Gg)** | **2006-2010 HFCs Cumulative Emissions (Gg) and $CO_2$-e Emissions (Gt $CO_2$-e)** | **2011-2015 HFCs Cumulative Emissions (Gg) and $CO_2$-e Emissions (Gt $CO_2$-e)** | **% change in cumulative emissions between 2006-2010 and 2011-2015** |
| **125** | $60.3 \pm 9.5$ | 131.3 (0.46) | 251.2 (0.88) | 91.7 |
| **134a** | $209.4 \pm 42.9$ | 736.6 (1.05) | 943.8 (1.35) | 28.1 |
| **143a** | $27.3 \pm 3.0$ | 89.2 (0.4) | 122.0 (0.55) | 36.9 |
| **32** | $31.9 \pm 14.4$ | 51.6 (0.04) | 125.2 (0.09) | 142.7 |
| Total | | **1009 (1.9)** | **1442 (2.9)** | **43** |

Even though global production and consumption of HCFCs in Article 5 countries was capped in
2013 these developing countries have substantially increased their usage of HCFCs (Montzka et al.,
2009) and are not required to phase-out potentially emissive consumption until 2040. As noted in Figure
2 the HCFCs show decreasing annual rates of growth with HCFC-142b declining by ~93% from 2007-
2015, HCFC-141b by 53% and HCFC-22 by 54%. This is reflected in the aggregated HCFCs global
$CO_2$-e emissions (shown as in Figure 6) exhibiting a small but steady decline from 2010-2015. In spite of
the 2013 cap on global production and consumption, the pace of decline has to some extent been
moderated by the increased emissions of HCFC-22 and HCFC-141b in article 5 countries. HCFC-22
remains the dominant refrigerant used in Article 5 countries and it has been estimated that approximately
1 million tonnes of HCFC-22 are currently in use in air conditioners operating worldwide (UNEP,
2016b). As the HFCs have replaced HCFCs, their aggregated emissions in 2015 have risen to a level in
$CO_2$-e that is 13% less than the aggregated emissions of the HCFCs.
Although there has been a shift in developed countries (non-Article 5) from HCFCs to high-GWP
HFCs (Lunt at al., 2015; Montzka et al., 2015) the consumption of both HCFCs and HFCs in Article 5
countries has substantially increased, most notably in China (Fang et al., 2012; Zhang et al., 2014; Su et
al., 2015; Velders et al., 2015). There has been a trend in recent years to move to refrigerant blends with





lower GWPs and in Japan in 2014 residential air-conditioners were switched to using HFC-32 (UNEP,
2016b).
HFCs-134a, -125, -143a and -32 are the principal components of all alternative substitutes for the
HCFCs and in the supplementary material (3) we examine the composition of the main refrigerant blends
to determine if there is evidence for a significant use of single component refrigerants. In general, we find
that the atmospheric mole fractions of HFCs -32, -125, and -143a are consistent with their release
predominantly as blends.

**5. Conclusions**
This study confirms that the Montreal Protocol and its amendments have been effective in
slowing the atmospheric accumulation of the HCFCs. If there had been no change in the emissions
growth rate, we find that based on our linear projection there would have been an additional $0.67 \pm 0.24$
Gt $CO_2$-e of aggregated HCFCs -22, -141b, -142b emitted to the atmosphere from 2010-2015. This
compares with the forward projections of 'business as usual' in the Velders et al., 2009 and WMO (2010)
scenario of an additional $0.71 \pm 0.25$ Gt $CO_2$-e and $0.66 \pm 0.24$ Gt $CO_2$-e, respectively, of total HCFCs
that would have been emitted during this same period. However, we also calculate that the aggregated
cumulative emissions of HCFCs-22, -141b, -142b, and -124 during the most recent 5-yr period (2011-
2015) are slightly larger (1.4%) than in the previous five years (2006-2010). This increase has likely been
driven by the substantial emissions of HCFCs-22 and -141b in Article 5 countries. HCFC-22 represents
about 79% of total global HCFC emissions. As shown in Figure 2, the annual HCFC-22 growth rate has
steadily declined since the introduction of the 2007 adjustment to the Montreal Protocol, yet global
emissions have tended to remain approximately constant with only a modest decline in emissions post-
2007 (see Figure 5). We also note the linkage between HCFC-22 and HFC-23, also a potent GHG, which
is an unavoidable by-product of HCFC-22 production, (Miller et al., 2010; Rigby et al., 2014). Therefore
further reductions in HCFC-22 production and consumption will benefit the efforts of the UNFCCC
Clean Development Mechanism (CDM) that mitigates HFC-23 emissions by voluntary incineration.
Since we are only two years beyond the 2013 cap on global production and consumption of HCFCs, it is
probably too early for our current observations through 2015 to show an accelerating phase-out for all
HCFCs, although HCFCs-142b and -124 have both recently undergone substantial declines in global
emissions.
Although global emissions of HFCs have increased throughout the course of this study,
Montzka et al., (2014) suggested that there may have been a shift to lower GWP refrigerant blends which
can account for the observed emissions. It is also noteworthy that the two HFCs with the largest
percentage changes in emissions in recent years are HFCs-125 and -32 (see Table 3), which implies a



trend towards blends containing these two refrigerants. However, we are unable to confirm the extent to
which other lower GWP blends (e.g. R404A and R507A) have been substituted with the possibility that
there has simply been a switch from one blend to another with similar GWPs. In terms of $CO_2$-e,
surprisingly HCFC-22 emissions alone contribute 0.45 Gt $CO_2$-e, about 14% more to climate change than
the four aggregated HFCs during 2011-2015. This could potentially be attributed to the continuing use of
HCFC-22 in existing refrigeration equipment with the consequence of large slowly leaking banks.

With regard to inventory-based HFC emissions estimates it is important to acknowledge that

attempting to quantify the release of individual HFCs to the atmosphere is complicated by the continuous
introduction of new blends many of which contain hydrocarbons and lower GWP refrigerants and their
substitution into older existing refrigeration equipment. This problem is further compounded by the lack
of information on the actual usage of the various blends in commercial and residential refrigeration,
coupled with the difficulty of quantifying emission magnitudes from the many banks and immediate
release as solvents and in foam blowing applications. Nevertheless, the atmospheric mole fractions
observed are consistent with emissions of HFCs in refrigerant blends, rather than substantial emissions
from single component refrigerants.

We find that the increase in HFC emissions from 2010 to 2015 has been more rapid than a linear

growth would imply. However, compared to this linear trend, the cumulative excess of emissions during
this period is 0.12±0.15 Gt $CO_2$-e, which is smaller than the deficit in HCFCs during the same time
frame. This suggests that the phase-down in HCFCs post-2010 has not coincided with an equivalent
increase in HFC emissions in $CO_2$-e terms. Compared to alternative projections by Velders et al. (2009),
our top-down estimates show a much more rapid decline in emissions of HCFCs, and slower increase in
HFC emissions. Therefore, similarly to our linear projection, where Velders et al. (2009) predicted a
relatively steady growth in $CO_2$-e emissions due to HCFCs and HFCs from 2005 – 2015, we find an
overall slowing of emissions post-2010.

Finally we note that national regulations to limit HFC use are already in place in the European

Union, Japan and the USA, and recently there has been an agreement to amend the Montreal Protocol to
further restrict HFC use beginning in 2019, (28[th] meeting of the parties to the Montreal Protocol, Kigali,
Rwanda, October 2016). By including HFCs, which have been shown to have a small, but non-zero
ozone-depletion potential (Hurwitz et al., 2015) into the Montreal Protocol, this has the benefit of further
regulating production and sales.






**Acknowledgements**

We specifically acknowledge the cooperation and efforts of the station operators (G. Spain, Mace Head, Ireland; R. Dickau, Trinidad Head, California; P. Sealy, Ragged Point, Barbados; NOAA officer-in-charge, Cape Matatula, American Samoa; S. Cleland, Cape Grim, Tasmania) at the core AGAGE stations and all other station managers and staff. The operation of the AGAGE stations was supported by the National Aeronautics and Space Administration (NASA, USA) (grants NAG5-12669, NNX07AE89G, NNX11AF17G and NNX16AC98G to MIT; grants NAG5-4023, NNX07AE87G, NNX07AF09G, NNX11AF15G and NNX11AF16G to SIO); the Department of the Energy and Climate Change (DECC, UK) (contract GA0201 to the University of Bristol); the National Oceanic and Atmospheric Administration (NOAA, USA) (contract RA133R09CN0062 in addition to the operations of American Samoa station), by the  Commonwealth Scientific and Industrial Research Organisation (CSIRO, Australia), the Bureau of Meteorology (Australia) and Refrigerant Reclaim Australia. M. Rigby is supported by a NERC Advanced Fellowship NE/I021365/1. We finally thank G. Velders and B. Xiang for supplying actual datasets from their publications.

**Data availability**

The entire ALE/GAGE/AGAGE data base comprising every calibrated measurement including pollution events is archived on the Carbon Dioxide Information and Analysis Center (CDIAC) at the U.S. Department of Energy, Oak Ridge National Laboratory.

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
