# Peer review of "Changing trends and emissions of hydrochlorofluorocarbons and their hydrofluorocarbon replacements."

_Atmospheric Chemistry and Physics, 2016_

## Referee Comment (RC1) · Anonymous Referee #2 · 12 Jan 2017

This manuscript provides updates on measurements of HCFCs and HFCs from a global sampling network that provide a global picture of the transition being made as a result of the Montreal Protocol. Results are provided and discussed in terms of atmospheric changes and inferred emission rates. Comparisons are made to emissions derived previously on a mass basis and are considered also on the basis of CO2-equivalent emissions for individual gases and for classes of gases. The paper presents high-quality measurement data that add to our understanding of recent atmospheric changes stemming from the Montreal Protocol. I found some sections in need of additional consideration before publication in ACP would be appropriate.

On uncertainties: It's not clear that the change derived for aggregate HCFC emissions

from 2010 to 2015 is accurately characterized as a decrease given the substantial overlap in the stated uncertainties. The two different estimates are 483 +/- 70 and 444 +/- 75 (this decrease is mentioned in multiple places in the text). Same point can be made for the 1.4% difference in cumulative emissions over the two five-year periods (lines 553-559). This needs more careful consideration and an accurate description. The "increase" in aggregate HFC emission values also need considering, as there is substantial overlap there too. I also find it surprising that the uncertainties on global values provided in Figure 1 and 2 aren't dependent on mole fraction or the number of sampling stations used to derive the values (2 sites in early years with the ADS and more sites recently with the updated Medusa instruments). Why isn't this observed? Were the early measurements from two sites much more precise?

On implications for compliance with the Montreal Protocol: The text in the abstract (lines 29-33) and on lines 584-595 can be read to suggest that usage of HCFC has increased after 2013 despite the global cap on production and consumption. Text on lines 589-591 suggests developing country emissions have increased in spite of the 2013 cap on production. These seem to be fairly significant statements with important implications but no evidence is supplied to back them up. I don't doubt that HCFC emissions and use increased prior to 2013 in developing countries, but what evidence suggests that use and emissions increased after 2013 from these countries?

On comparisons with emission estimates presented previously: Emission estimates for many gases and many sources (Figure 3). It's great to see the authors provide emission estimates from previous work for comparison of derived magnitudes and trends. Although I'm not sure it is surprising that CO2-e emissions of HCFC-22 are larger than the four HFCs, has this not been obvious from earlier work and WMO assessments? Regarding figure 3, it would be more useful for the reader if it were clear which results were derived independently from the AGAGE data (from different observations and model), which were derived independently from the AGAGE 12-box model but with AGAGE data, and which were derived from inventories (e.g., what are Velders et al.,

results derived from?). Also, a quick look at the Montzka et al., 2015 paper shows emissions derived and presented for HCFCs and HFCs for many years, not just 2012 (only 2012 results are plotted in this manuscript). This comes across as a bit misleading, but more importantly, the authors miss a significant opportunity to determine if the two measurement networks provide similar conclusions regarding the unusual inter-annual changes in emissions for these gases (particularly the uneven changes for HCFCs).

Unusual insertions in the text: 1) The first mention of HFC-23 is in the conclusion section. This seems out of place and, I'd suggest, inappropriate given that none of the information provided about HFC-23 is derived from data or analyses of observations presented in this manuscript and the points made aren't closely relevant for this manuscript. 2) The discussion of HFCs being released predominantly in blends seems out of place and unusual. This is a straightforward conclusion based on uses of these gases by industry and it is not clear how the atmospheric data add to this discussion. There is a related point made in the conclusion about results not agreeing with some from Montzka et al (2014?), but there is no indication given as to the reason for this difference. Is it because the derived emissions disagree or is it because more information was brought to the analysis in the present manuscript than was available previously that refines our knowledge?

Details: Citation seems important but is lacking on line 50.

Lines 61-65. Have no HFC results been reported by NOAA since 2004?

Precisions are quoted on lines 151-154 as single numbers, but I would guess that they have changed over time with different instruments and as atmospheric mole fractions have increased from v small levels. Does typical = median?

Results and Discussion: How comparable are the model output mole fractions to the actual results? No indication of this is presented or mentioned.

Are growth rates quoted (line 237-238) based on some time interval, or just the measured change during 2015?

Line 309, reconsider text. HCFC-141b growth rate isn't reported before 1998, so it doesn't seem accurate to suggest that emissions peaked in that year.

Consider units on increasing emission rates as per yr per yr.

Line 499 and 561-562. I believe this is correct only if you refer to relative rates of increase.

Line 565. "emissions of HCFC-22 represent 79% of the global cumulative HCFC burden..." doesn't make sense. Is the percentage relating to mole fractions or emissions?

WMO reports are appropriately cited by lead coauthor names; consider doing that as recommended in the reports.
* * *

---

## Referee Comment (RC2) · Anonymous Referee #1 · 12 Jan 2017

This is undoubtedly a very good and timely work updating trends and emissions of 8 trace gases of importance to stratospheric ozone depletion and/or global warming. I consider it to be of sufficient quality and importance to be published in ACP. There are however some concerns that need to be addressed beforehand. Two of them stand out:

1. In section 4.1. atmospheric mole fractions and growth rates are discussed but there is no attempt to compare any of these with existing published data, e.g. from the multiple articles cited in the introduction. There is only a discussion for emissions, which leaves the reader in the dark as to whether emission estimates might agree for the wrong reasons (e.g. model differences). 2. As also detailed in the comments below, some of the uncertainties presented are not very well explained. It is for instance unclear to the reader why global atmospheric mole fractions and growth rates are so much less uncertain than global emissions. A brief explanation of the major contributing factors to these uncertainties would help.

In addition there are a substantial number of (partly equally important) comments on specific parts of the manuscript that require attention prior to publication: 1. Line 1-2 and 19-20. Consider including "HCFCs" and "HFCs" in the title or "hydrochlorofluorcarbons" and "hydrofluorcarbons" in the abstract.

2. Line 27-28. This reduction is not significant even within 1 sigma, so it is rather questionable to call it a reduction. And how can "all quoted uncertainties in this paper" be 1 sigma if it is stated later that some represent a complicated mix of "uncertainties due to the observations, the prior and the current best-estimate lifetimes of these compound..."? A clear and ideally brief explanation of what different uncertainties stand for and how these were calculated is needed.

3. Line 29. The second "HCFC" is not needed.

4. Line 33. "increasing annual growth rates" presumably refers to mole fractions, not emissions? I find it somewhat surprising that the abstract does not include any numbers for mole fractions, especially since the first two words of the title are "Changing trends". 5. Line 42-50. The first paragraph is not backed up with any references whatsoever.

6. Line 51. It is worth noting why they have been introduced as replacements for the HCFCs and CFCs.

7. Line 52-53. I recommend citing Table 1 here.

8. Line 68. Please name these stations or cite Table 2 here. I also find it rather surprising that no station data is shown anywhere in the manuscript or the supplement. Perhaps it would be more appropriate to add "global" to the title if regional trends and

emissions are not covered?

9. Line 74. These are not just lifetimes but steady-state atmospheric lifetimes.

10. Line 77-78. The most recent estimates of ODPs and GWPs are published in the last WMO Ozone Assessment (2014) as well as the last IPCC report (2013) and I urge the authors to cite the appropriate chapters from these assessments as well as to not use outdated numbers in their calculations.

11. Line 81. Please explain what is meant by "tuning".

12. Line 97-108. Again, none of these statements is backed up by any reference.

13. Line 109. Why are these "non-zero" ODPs given as "0" in Table 1?

14. Line 117-120. I suggest changing the title of this paragraph to "AGAGE sites" or similar, and to mention "coordinates".

15. Line 146. Please explain the technique of filling standards "cryogenically". Have any differences been observed between standards filled with the two concurrent techniques?

16. Line 176-178. This is rather opaque. Please explain the methodology of calculating these uncertainties.

17. Line 232-234. Please clarify whether "2015" means the end of that year or the annual average.

18. Line 235-238. I suggest moving this paragraph to after the end of the HCFC mole fraction and growth rate discussion.

19. Line 308-309. I disagree. From the data shown in Figure 1 and 2 it is not clear whether HCFC-141b growth rates where at a maximum in 1998 as this is where the data starts. It might also be worth mentioning somewhere in this manuscript that data exists for earlier years but is not being focused on (and why).

20. Line 320. I find this nomenclature confusing as there is only one supplementary material file.

21. Line324. Surely there are some EDGAR emissions for HCFC-22. Why are these not included or discussed?

22. Line 325. How can emissions be derived from 1995 onwards using global mole fractions that start in 1998?

23. Line 335-336. 63 $\pm$ 6 Gg/yr and 31 $\pm$ 6 Gg/yr are not similar. Also, the minimum occurred during 2004-2005, not the decline.

24. Line 342-343. This is quite a striking difference. Can the authors offer any explanation?

25. Line 344. Please check the number for HCFC-141b.

26. Line 388-398. The quality of this figure is much worse than the others.

27. Line 477. How do these emissions compare with previously published estimates in O'Doherty et al., 2009 and 2014?

28. Line 480. Why is HFC-125 listed first here and in the figure?

29. Line 495-497. This should probably be "in agreement with" instead of "in spite of".

30. Line 499. Please clarify whether "most rapid" refers to relative or absolute changes.

31. Line 514. Have the authors considered that Velders et al., 2009 also included HFC-152a, HFC-245fa, and HFC-365mfc in their calculations?

32. Line 550. Why was Velders et al., 2009 data "rescaled"?

33. Line 555-556. This sentence seems to disagree with the previous one.

34. Line 591. Can the authors present any evidence for this claim?

35. Line 595. It would help to include the starting point of that rise in order to illustrate

it.

36. Line 598. "at al."

37. Line 604-607. This is implied to be a main point of the manuscript as it is included in the abstract and the conclusions (lines 635-637 and 647-649). I strongly recommend either moving this analysis to the main manuscript or removing the respective statements.

38. Line 620-623. This is quite surprising. The HCFC-22 growth rate has dropped from around 8 pmol/mol to less than 4 pmol/mol in that period, yet the global emissions have not been affected much? I think many readers would be interested in an explanation for this apparent disconnect.

39. Line 625-626. I thought the CDM had expired?

40. Line 650-652. Please explain why only above-linear HFC growth should be related to a deficit in HCFC emissions.

41. Line 654-658. Why are the HFC results only compared to Velders et al. (2009) and not the updated and improved projections from Velders et al. (2015)?

---

## Author Comment (AC1) · 1 Mar 2017

Responses to Referees We thank both referees for their diligence in reviewing our paper and the very constructive and substantive comments which have improved the paper.

REFEREE #1

Atmos. Chem. Phys. Discuss., Doi:10.5194/acp-2016-977-RC2, 2017 [©] Author(s) 2017. CC-BY 3.0 License. Interactive comment on "Changing trends and emissions of hydrochlorofluorocarbons and their hydrofluorocarbon replacements" by Peter G. Simmonds et al.

This is undoubtedly a very good and timely work updating trends and emissions of 8 trace gases of importance to stratospheric ozone depletion and/or global warming. I consider it to be of sufficient quality and importance to be published in ACP. There are however some concerns that need to be addressed beforehand. Two of them stand out: 1. In section 4.1. atmospheric mole fractions and growth rates are discussed but there is no attempt to compare any of these with existing published data, e.g. from the multiple articles cited in the introduction. There is only a discussion for emissions, which leaves the reader in the dark as to whether emission estimates might agree for the wrong reasons (e.g. model differences). 2. As also detailed in the comments below, some of the uncertainties presented are not very well explained. It is for instance unclear to the reader why global atmospheric mole fractions and growth rates are so much less uncertain than global emissions. A brief explanation of the major contributing factors to these uncertainties would help. In addition there are a substantial number of (partly equally important) comments on specific parts of the manuscript that require attention prior to publication:

1. We have added the following text regarding mole fractions to the abstract. "Global mean mole fractions of HCFC-22, -141b, and -142b have increased throughout the observation period reaching 234, 24.3 and 22.4 pmol mol-1, respectively in 2015. HCFC-124 reached a maximum global mean mole fraction of 1.48 pmol mol-1 in 2007 and has since declined by 23% to 1.14 pmol mol-1 in 2015. The HFCs all show increasing global mean mole fractions. In 2015 the global mean mole fractions (pmol mol-1) were 83.3 (HFC-134a), 18.4 (HFC-125), 17.7 (HFC-143a) and 10.5 (HFC-32)."

We have also added a brief comparison of our mole fractions with the most recent published data (Montzka et al., 2015) which used observations from the independent NOAA network. (See Section 4.1). In addition, the Supplementary Material includes a section on comparison of NOAA flask and AGAGE in situ HCFC measurements at common sites. However, some of our cited references refer specifically to regional mixing

ratios and emissions and therefore are not readily comparable with our global results. "To compare AGAGE HCFCs results with the recent measurements reported by NOAA (Montzka et al., 2015), we list NOAA 2012 global mean mole fractions (pmol mol-1) and in parenthesis the corresponding AGAGE 2012 global mixing ratios: HCFC-22, 218.2 (219.4); HCFC-141b, 22.3 (22.5) and HCFC-142b, 21.5 (21.9). In addition, the Supplementary Material includes a section on comparison of NOAA flask and AGAGE in situ HCFC measurements at common sites, which is summarised here as percentage differences (NOAA/AGAGE-1)*100: HCFC-22, -0.3±0.3%; HCFC-141b, -0.6±0.5%; and HCFC-142b, -2.6±0.5%. These comparisons between the two independent observing networks are generally in good agreement."

and "Montzka et al., 2015, reported 2012 HFCs mean mole fractions (pmol mol-1) which we compare with our 2012 global mixing ratios in parenthesis. HFC-134a, 67.5 (67.7); HFC-143a, 12.3 (13.4); HFC-125, 11.4 (12.1) and HFC-32, 5.1 (6.3). The Supplementary Material also includes a section on comparisons of NOAA flask and AGAGE in situ HFC measurements at common sites, which is summarised here as percentage differences (NOAA/AGAGE-1)*100: HFC-134a, 0.1±0.5%; HFC-143a, -7.7±0.8%; HFC-125, -5.4±0.8%; and HFC-32, -9±2%."

Detailed emissions estimates comparisons with results from some of the many cited publications are shown in Figures 3(a-d) and 4(a-d) with additional discussion in sections 4.2.1 and 4.2.2. 2. We have also added some more explanation about how uncertainties were calculated, as detailed below (note that full details of the uncertainty quantification are provided in Rigby et al. (2014)). For the reviewer's comment regarding the relative uncertainty in the mole fractions versus emissions (notwithstanding the fact that it is difficult to compare relative uncertainties in these two different quantities with different units), the mole fractions and growth rates are less uncertain than emissions rates because: a) the observations directly measure mole fractions, whereas emissions must be inferred indirectly using the model; b) in particular, emissions uncertainties also include a contribution from the assumed lifetime, an uncertainty to

which global mole fractions are not subject.

1. Line 1-2 and 19-20. Consider including "HCFCs" and "HFCs" in the title or "hydrochlorofluorcarbons" and "hydrofluorcarbons" in the abstract. HCFCs and HFCs have been added to the title

2. Line 27-28. This reduction is not significant even within 1 sigma, so it is rather questionable to call it a reduction. And how can "all quoted uncertainties in this paper" be 1 sigma if it is stated later that some represent a complicated mix of "uncertainties due to the observations, the prior and the current best-estimate lifetimes of these compound..."? A clear and ideally brief explanation of what different uncertainties stand for and how these were calculated is needed.

We have changed this sentence to say: "We find that this change has coincided with a stabilisation, or moderate reduction, in global emissions of the four HCFCs with aggregated global emissions in 2015 of 449 $\pm$ 75 Gg/yr, in $CO_2$ equivalent units ($CO_2$ e) 0.76 $\pm$ 0.1 Gt/yr, compared with 483 $\pm$ 70 Gg/yr (0.82 $\pm$ 0.1 Gt/yr $CO_2$ e) in 2010 (uncertainties are 1-sigma throughout this paper)."

We do not completely follow the reviewer's comment regarding the "complicated mix" of uncertainties as it applies here. This sentence is simply saying that quoted uncertainties, even where they represent those calculated from a variety of contributing sources, are 1-sigma. To very briefly clarify how the uncertainties were calculated, we have replaced the sentence at the end of Section 3, which was originally on line 222 – 224, to say: "Our methodology calculates uncertainties in each derived quantity such as global emissions, mole fractions or growth rates as described in Rigby et al. (2014). Briefly, uncertainties in each quantity comprise contributions from the measurement uncertainty and an estimate of the model representation error (which was taken to be equal to the monthly baseline variability), which are propagated through the inversion to each derived parameter. Since these uncertainties only represent "unbiased" or random sources of error, we then add the influence of potential biases due to the calibration scale and uncertain lifetime estimates. The uncertainty on global mean mole fractions includes the contribution of errors in the calibration scale, and uncertainties in derived emissions include contributions from the calibration scale and the lifetime uncertainty. The total uncertainty is calculated from these random and bias terms from a Monte Carlo ensemble in which each member has a perturbed value of each type of uncertainty."

3. Line 29. The second "HCFC" is not needed. Second HCFC has been removed

4. Line 33. "increasing annual growth rates" presumably refers to mole fractions, not emissions? I find it somewhat surprising that the abstract does not include any numbers for mole fractions, especially since the first two words of the title are "Changing trends".

We have added the following sentences to the abstract.

"Global mean mole fractions of HCFC-22, -141b, and -142b have increased throughout the observation period reaching 234, 24.3 and 22.4 pmol mol-1, respectively in 2015. HCFC-124 reached a maximum global mean mole fraction of 1.48 pmol mol-1 in 2007 and has since declined by 23% to 1.14 pmol mol-1 in 2015. The HFCs all show increasing global mean mole fractions. In 2015 the global mean mole fractions (pmol mol-1) were HFC-134a (83.3), HFC-125 (18.4), HFC-143a (17.7) and HFC-32 (10.5)."

5. Line 42-50. The first paragraph is not backed up with any references whatsoever. Line 44. New reference has been added. consumption grew rapidly in developed countries until the mid-1990s (AFEAS, 2016).

Line 48. Text modified and new reference has been added. "Subsequently the 2007 adjustments to the Montreal Protocol required an accelerated phase-out of emissive uses of HCFCs in both "non-Article 5" developed and "Article 5" developing countries, with a 2004 cap on production and consumption for non-Article 5 countries and a 2013 global cap on production and consumption (UNEP, 2016a)."

[Figure]

Lines 49 & 50. Text corrected and reference added. Historically, HCFCs-22, -141b and -142b account for >90% of the total consumption of all HCFCs in non-Article 5 countries (AFEAS, 2016).

6. Line 51. It is worth noting why they have been introduced as replacements for the HCFCs and CFCs.

Line 61 (not 51) We consider that this is covered adequately in the paragraph above (lines 42 to 50) and there is no need to repeat it here.

7. Line 52-53. I recommend citing Table 1 here.

Table 1 has been moved as requested

8. Line 68. Please name these stations or cite Table 2 here. I also find it rather surprising that no station data is shown anywhere in the manuscript or the supplement. Perhaps it would be more appropriate to add "global" to the title if regional trends and emissions are not covered?

All of the individual AGAGE station datasets are available on the CDIAC website which we noted on lines 686-688. It would have considerably lengthened the paper if we had included all of these datasets. Table 2 has been moved as requested and "Global" has been added to the title.

9. Line 74. These are not just lifetimes but steady-state atmospheric lifetimes.

Changed to Steady-state Atmospheric Lifetimes

10. Line 77-78. The most recent estimates of ODPs and GWPs are published in the last WMO Ozone Assessment (2014) as well as the last IPCC report (2013) and I urge the authors to cite the appropriate chapters from these assessments as well as to not use outdated numbers in their calculations.

We respectfully disagree with this comment. In the context of this paper, the important ODP and GWP values are those used by regulators to place controls on the title subnone

stances. This means that the ODP values are those of the Montreal Protocol (UNEP, 2016a) and the GWP values are those of the 4th Assessment of IPCC (as cited). If the reviewer cares to follow the history of GWPs through the IPCC Assessments, he/she will see that they vary in time and that "most recent" is not necessarily most accurate or "best". What is important is to reference the source of the numbers, which has been done.

11. Line 81. Please explain what is meant by "tuning".

This is somewhat of an over-simplification. We have re-written this sentence as: "We combine these observations with a 2-dimensional (12-box) atmospheric chemical transport model whose circulation is based on meteorological climatology, which has been adjusted in inverse modelling studies to provide improved agreements with global distributions of reactive and stable trace gases (Cunnold et al., 1983; Rigby et al., 2011)."

12. Line 97-108. Again, none of these statements is backed up by any reference.

Added reference to line 102, "extinguishers and as a component of sterilant mixtures (Midgley & McCulloch, 1999)". Added reference to line 108, "blends used in commercial refrigeration and in some air conditioning applications (Ashford et al., 2004)". 13. Line 109. Why are these "non-zero" ODPs given as "0" in Table 1? Because HFCs were not included in the Montreal Protocol before 2016 by virtue of the fact that they do not chemically deplete ozone (hence, by definition their ODPs are zero). The Hurwitz values are second order, from the effects of temperature & dynamics on stratospheric ozone (in which case, carbon dioxide and methane are also ODS). During 2016, HFCs were introduced into the Montreal Protocol as the most convenient means of regulating their climate impacts but this had nothing to do with ozone depletion.

14. Line 117-120. I suggest changing the title of this paragraph to "AGAGE sites" or similar, and to mention "coordinates".

Changed to 'AGAGE sites' as requested, co-ordinates are listed in Table 2.

15. Line 146. Please explain the technique of filling standards "cryogenically". Have any differences been observed between standards filled with the two concurrent techniques?

Following sentences have been added to the paper. "This method of cryogenically collecting large volumes of ambient air is the same as that used for collecting air for the Cape Grim Air Archive (Langenfelds et al., 1996). Briefly, the evacuated cylinder is partially (about 75%) immersed in a bath of liquid nitrogen, and ambient air flow into the cylinder is assisted by a small, clean, metal bellows pump (Metal Bellows Corp, MB-118E, P/N 31185). . The flow rate of the air and the elapsed time determine the volume of air collected. Measurements of many atmospheric trace species in air samples collected in this manner show that the trace gas composition of the air is well preserved (Langenfelds et al., 1996)."

16. Line 176-178. This is rather opaque. Please explain the methodology of calculating these uncertainties.

We have expanded and improved Section 2.3 on Calibration Scales with a clearer description of our methodology. "The estimated accuracies of the calibration scales for the various HCFCs and HFCs are reported below, and more detailed discussion of the measurement techniques and calibration procedures are reported elsewhere (Miller et al., 2008; O'Doherty et al., 2009; Mühle et al., 2010). As noted in the preceding section, these AGAGE HCFC and HFC measurements are reported relative to Scripps Institution of Oceanography (SIO) and University of Bristol (UB) primary calibration scales: SIO-05 (HCFCs-22, -141b, -142b, and HFC-134a); UB 98 (HCFC-124); SIO-07 (HFCs -143a, and -32); and SIO-14 (HFC-125). SIO calibration scales are defined by suites of standard gases prepared by diluting gravimetrically prepared analyte mixtures in $N_2O$ to near-ambient levels in synthetic air (Prinn et al., 2000; Miller et al., 2008) and UB calibration scales are defined by similar dilutions of commercially prepared (Linde

Gases UK) analyte mixtures (O'Doherty et al., 2004). Results are reported as dry gas mole fractions in pmol mol-1 (or parts-per-trillion – ppt). The absolute accuracies of these primary standard scales are difficult to assess because they are vulnerable to systematic effects that are difficult to quantify or may not even be identified. This is why the use of traceable calibration scales that are tied to a maintained set of specific calibration mixtures is of paramount importance in the measurement of atmospheric composition change. Combining known uncertainties such as measurement and prop-agation errors and quoted reagent purities generally yields lower uncertainties than are supported by comparisons among independent calibration scales (Hall et al., 2014). Furthermore, some systematic uncertainties may be normally distributed, while others like reagent purity are skewed in only one direction. Estimates of absolute accuracy are nevertheless needed for interpretive modelling applications, and in this work they are liberally estimated at: 2% for HCFC-22, -141b, and -142b; 10% for HCFC-124; 1.5% for HFC-134a; and 3% for HFC-125, -143a, and -32."

17. Line 232-234. Please clarify whether "2015" means the end of that year or the annual average.

The sentence has been amended to:- "to 1.14 pmol mol-1 in 2015."

18. Line 235-238. I suggest moving this paragraph to after the end of the HCFC mole fraction and growth rate discussion.

Agreed, paragraph has been moved.

19. Line 308-309. I disagree. From the data shown in Figure 1 and 2 it is not clear whether HCFC-141b growth rates where at a maximum in 1998 as this is where the data starts. It might also be worth mentioning somewhere in this manuscript that data exists for earlier years but is not being focused on (and why).

We apologise for this error. Our plots had been cut off at 1998, as this is the year in which the time series begin for several compounds. We have now extended these plots

back to 1995, when in situ data began for some of these species.

20. Line 320. I find this nomenclature confusing as there is only one supplementary material file. The Supplementary file is now a single file.

21. Line324. Surely there are some EDGAR emissions for HCFC-22. Why are these not included or discussed?

EDGAR does not have emissions for HCFC-22, only 141b and 142b.

22. Line 325. How can emissions be derived from 1995 onwards using global mole fractions that start in 1998?

We apologise for this error as noted in Point 19 above.

23. Line 335-336. 63±6 Gg/yr and 31±6 Gg/yr are not similar. Also, the minimum occurred during 2004-2005, not the decline.

Text has been clarified as follows:- "These two HCFCs have exhibited several fluctuations in emissions with maxima around 2000 of 63 ± 6 Gg/yr (HCFC-141b) and 32 ± 5 Gg/yr (HCFC-142b) followed by minima of 46 ± 7 Gg/yr (HCFC-141b) and 29 ± 4 Gg/yr (HCFC-142b) in 2003-2005." 24. Line 342-343. This is quite a striking difference. Can the authors offer any explanation? This statement about EDGAR is simply a statement of fact. We cannot account for the accuracy or inaccuracy of the EDGAR reported emissions.

25. Line 344. Please check the number for HCFC-141b. We thank the reviewer for noting this error. HCFC-141b has been corrected to 0.5%.

26. Line 388-398. The quality of this figure is much worse than the others.

Clarity of Figure 3d has been improved.

27. Line 477. How do these emissions compare with previously published estimates in O'Doherty et al., 2009 and 2014?

We have added expanded the discussion in Section 4.2.2 as follows:- "In contrast to the HCFCs, estimates of HFCs emissions fluxes shown in Figure 4 (a-d) (see Supplementary Material for actual values) have increased over the entire observational record, reaching maxima in 2015 of 209 $\pm$ 43 Gg/yr (HFC-134a), 60 $\pm$ 10 Gg/yr (HFC-125), 31 $\pm$ 14 Gg/yr (HFC-32) and 27 $\pm$ 3.0 Gg/yr (HFC-143a). These estimates are updates for HFC-32 and HFC-143a emissions reported in O'Doherty et al. (2014) and updates for HFC emissions reported in Rigby et al. (2014). The most recent HFCs emissions estimated from NOAA data (Montzka et al., 2015) are in most cases in close agreement with our emission estimates from AGAGE data and generally within our uncertainties. We include in Figure 4(b) previously published HFC-125 global emissions (O'Doherty et al., 2009) increasing from 6.1 Gg/yr (1998) to 20.5 Gg/yr (2007). This compares well with our current emission estimates of 5.5 to 21.8 Gg/yr (1998-2007)." Also HFC-125 data from O'Doherty et al., 2009 has been plotted in Figure 4b for comparison.

28. Line 480. Why is HFC-125 listed first here and in the figure?

Order has been repositioned. (134a,125,32,143a)

29. Line 495-497. This should probably be "in agreement with" instead of "in spite of".

Agreed. Changed to "in agreement with"

30. Line 499. Please clarify whether "most rapid" refers to relative or absolute changes.

Changed text to reflect relative changes.

31. Line 514. Have the authors considered that Velders et al., 2009 also included HFC-152a, HFC-245fa, and HFC-365mfc in their calculations?

We did note that Velders. 2009 had included other HFCs (-152a, -245fa and -365mfc) in their publication. This paper was intended to be a report on the major HCFCs and HFCs and we decided to omit the minor HFCs, since AGAGE measurements of other HFCs have been reported previously. HFC-152a (Simmonds et al., 2015); HFC-245fa and HFC-365mfc (Vollmer at al., 2011). We have added the following sentence to the

text with respect to this latter reference. "and global emissions of the minor HFCs -245fa and -365mfc have been reported in Vollmer et al., (2011)."

32. Line 550. Why was Velders et al., 2009 data "rescaled"?

This sentence has been added to the notes in Figure 6. "We have adjusted the start date of the Velders et al, 2009 emission trends to 2010 in alignment with the observationally derived value."

33. Line 555-556. This sentence seems to disagree with the previous one.

The first sentence applies to changes within each of the time periods and the second sentence refers to changes between the periods.

34. Line 591. Can the authors present any evidence for this claim?

The following reference has been added. "…..moderated by the increased emissions of HCFC-22 and HCFC-141b in article 5 countries (Carpenter & Reimann, 2014)."

35. Line 595. It would help to include the starting point of that rise in order to illustrate it.

This sentence has been re-phrased. "As the HFCs are replacing HCFCs, their aggregated emissions in 2015 of 327 Gg/yr have not yet reached the total HCFCs aggregated emissions of 449 Gg/yr, (see Table 3)."

36. Line 598. "at al."

Changed to "et al" (We thank the reviewer for spotting this "typo").

37. Line 604-607. This is implied to be a main point of the manuscript as it is included in the abstract and the conclusions (lines 635-637 and 647-649). I strongly recommend either moving this analysis to the main manuscript or removing the respective statements.

This is only one of the points we make in the paper. These statements are an integral

part of the analysis in the supplementary material, where we attempt to reconcile HFCs emissions with the composition of the known blends. We would prefer to leave this section in place rather than shifting it from the Supplementary Section.

38. Line 620-623. This is quite surprising. The HCFC-22 growth rate has dropped from around 8 pmol/mol to less than 4 pmol/mol in that period, yet the global emissions have not been affected much? I think many readers would be interested in an explanation for this apparent disconnect.

This is a mathematical consequence of the prevailing atmospheric mixing ratio and the atmospheric lifetime. The statement refers to a decline in the rate of growth of concentration which is perfectly compatible with a constant rate of emission.

39. Line 625-626. I thought the CDM had expired?

This is an accurate statement about the CDM which is still in existence but closed to new HFC-23 projects but continuing to honour previous obligations. New HCFC-22 producers are encouraged to mitigate HFC-23 emissions voluntarily.

40. Line 650-652. Please explain why only above-linear HFC growth should be related to a deficit in HCFC emissions.

Text has been changed to try and clarify our hypothesis. "We find that the increase in HFC emissions from 2010 to 2015 has been more rapid than the linear projection of growth, shown in Figure 6, would imply. This may provide some insight into the relative phase in of HFCs and phase out of HCFCs. Compared to this linear trend, the cumulative excess of HFCs emissions during this period is 0.12±0.15 Gt CO2-e, which is smaller than the deficit (-0.67±0.24 Gt CO2-e) in HCFCs over the same time frame."

41. Line 654-658. Why are the HFC results only compared to Velders et al. (2009) and not the updated and improved projections from Velders et al. (2015)?

The following text has been added. "Because Velders et al., 2009 and 2015 are similar over the 1995 to 2015 timeframe we opted to make the comparison with the most

widely quoted 2009 paper, also noting that the 2015 paper includes an assimilation of atmospheric observed abundances (Velders et al., 2015)."

Text on original lines 484-486 have been modified. "Published HFC-134a emissions estimates by Velders et al., (2009) are in close agreement with the results from this work. For HFC-32 the Velders et al. results agree within the uncertainties of our estimates with the exception of the early period from 1995-2002. Post-2012, Velders et al. HFC-125 and HFC-143a projections begin to diverge substantially from our emission estimates."

---

## Author Comment (AC2) · 1 Mar 2017

Responses to Referees We thank both referees for their diligence in reviewing our paper and the very constructive and substantive comments which have improved the paper.

REFEREE #2

Atmos. Chem. Phys. Discuss., doi:10.5194/acp-2016-977-RC1, 2017 © Author(s) 2017. CC-BY 3.0 License. Interactive comment on "Changing trends and emissions of hydrochlorofluorocarbons and their hydrofluorocarbon replacements" by Peter G. Simmonds et al. Anonymous Referee #2
This manuscript provides updates on measurements of HCFCs and HFCs from a global sampling network that provide a global picture of the transition being made as a result of the Montreal Protocol. Results are provided and discussed in terms of atmospheric changes and inferred emission rates. Comparisons are made to emissions derived previously on a mass basis and are considered also on the basis of CO2-equivalent emissions for individual gases and for classes of gases. The paper presents high-quality measurement data that add to our understanding of recent atmospheric changes stemming from the Montreal Protocol. I found some sections in need of additional consideration before publication in ACP would be appropriate.

On uncertainties: It's not clear that the change derived for aggregate HCFC emissions from 2010 to 2015 is accurately characterized as a decrease given the substantial overlap in the stated uncertainties. The two different estimates are 483 +/- 70 and 444 +/- 75 (this decrease is mentioned in multiple places in the text). Same point can be made for the 1.4% difference in cumulative emissions over the two five-year periods (lines 553-559). This needs more careful consideration and an accurate description. The "increase" in aggregate HFC emission values also need considering, as there is substantial overlap there too. I also find it surprising that the uncertainties on global values provided in Figure 1 and 2 aren't dependent on mole fraction or the number of sampling stations used to derive the values (2 sites in early years with the ADS and more sites recently with the updated Medusa instruments). Why isn't this observed? Were the early measurements from two sites much more precise?

We have modified the relevant sentences to say: "We find that this change has coincided with a stabilisation, or moderate reduction, in global emissions of the four HCFCs. . . . . . . . . . . . ." and- "Aggregating the four HCFCs we observe that there is an equal contribution of 3.9 Gt CO2-e over the two 5-yr periods, implying a stabilization of the cumulative emissions."

I also find it surprising that the uncertainties on global values provided in Figure 1 and 2 aren't dependent on mole fraction or the number of sampling stations used to derive

the values (2 sites in early years with the ADS and more sites recently with the updated Medusa instruments). Why isn't this observed? Were the early measurements from two sites much more precise?

There is a small decrease in the relative uncertainty in the global mole fraction as the number of stations increases (e.g. for HFC-134a the uncertainty in the global mole fraction is 2.3% in 1995, dropping to around 1.5% in 2014). However, in our inversion the terms involving global mole fractions are strongly constrained using only two stations, so the difference is not dramatic when the number of stations is increased. This is most likely because the inversion sees that the model provides a constraint on the latitudinal gradient, and therefore information on the mole fraction in any box can be propagated through to a global average (it is important to note that our global averages are from the model into which data haves been assimilated, rather than a purely "data" driven average). It is possible that our global mole fraction estimates are somewhat over-confident, because, in this inversion, we have not accounted for the potential systematic model errors that would confound this propagation of information around the globe (e.g. if there were errors in the inter-hemispheric exchange rate, then global estimates based on Mace Head alone would also be erroneous). However, we would argue that, for this paper, these factors are relatively minor, and such uncertainties would not change the outcomes. In future, our inverse modelling framework will be modified to more fully account for potential systematic model errors, but it would require significant further work (note also that almost no existing inverse modelling schemes account for such uncertainties).

On implications for compliance with the Montreal Protocol: The text in the abstract (lines 29-33) and on lines 584-595 can be read to suggest that usage of HCFC has increased after 2013 despite the global cap on production and consumption. Text on lines 589-591 suggests developing country emissions have increased in spite of the 2013 cap on production. These seem to be fairly significant statements with important implications but no evidence is supplied to back them up. I don't doubt that HCFC

emissions and use increased prior to 2013 in developing countries, but what evidence suggests that use and emissions increased after 2013 from these countries?

There must be some confusion here as the text does not say what the referee is suggesting. Supplementary material shows how emissions in any year are not equal to consumption and production in that year. The substances are emitted from equipment that uses them, either during operation or when the equipment is eventually scrapped. Thus an increase in emissions is not indicative of a change in production or consumption and, over a period of several years, it is possible that emissions will exceed consumption significantly. As for the Article 5 emissions, it's this paper that's drawing the conclusion. The increase in emissions (that are mainly from China) are consistent with the projections beyond 2013 in Wan et al., (2009), Li et al., (2011), Fang et al., (2012) and Carpenter & Reimann et al. (2014).

On comparisons with emission estimates presented previously: Emission estimates for many gases and many sources (Figure 3). It's great to see the authors provide emission estimates from previous work for comparison of derived magnitudes and trends. Although I'm not sure it is surprising that CO2-e emissions of HCFC-22 are larger than the four HFCs, has this not been obvious from earlier work and WMO assessments? Regarding figure 3, it would be more useful for the reader if it were clear which results were derived independently from the AGAGE data (from different observations and model), which were derived independently from the AGAGE 12-box model but with AGAGE data, and which were derived from inventories (e.g., what are Velders et al., results derived from?).

We agree that this is not particularly surprising and "surprising" has been removed from the text. We have addressed the issue of which datasets and models are used by adding the following text in Section 4.2.1.1. "It should be noted that Montzka used a 3-box model and NOAA data, while Saikawa and Xiang used independent 3-D atmospheric chemistry-transport models with NOAA and AGAGE+NOAA data, respectively." Velders results are compiled from inventory sources but also take into account atmospheric observations.

Also, a quick look at the Montzka et al., 2015 paper shows emissions derived and presented for HCFCs and HFCs for many years, not just 2012 (only 2012 results are plotted in this manuscript). This comes across as a bit misleading, but more importantly, the authors miss a significant opportunity to determine if the two measurement networks provide similar conclusions regarding the unusual inter-annual changes in emissions for these gases (particularly the uneven changes for HCFCs).

With regard to the Montska 2015 paper. We reported an emission for 2012, as this was the only data listed in the text of the paper and it would have been inaccurate to try and read values for other years off the figures. However, Dr Montzka has kindly supplied us with a table of data for the other years (including recent revisions) which we have now included in the appropriate figures in our paper. In addition, we have included in the Supplementary Material a section on a comparison of NOAA flask and AGAGE in situ HCFC and HFC measurements at common sites, which is summarised here as percentage differences (NOAA/AGAGE-1)*100: HCFC-22, -0.3±0.3%; HCFC-141b, -0.6±0.5%; and HCFC-142b, -2.6±0.5%. It is quite clear from this co-plot of the AGAGE and NOAA data that variations in the HCFCs trends are well matched. Modified text as follows:- "Montzka et al., (2015), using an independent sampling network, also provided emissions estimates for HCFCs-141b and -142b which are included in Figures 3 b,c and agree within the uncertainties of our estimates with similar fluctuations."

Unusual insertions in the text:

(1) The first mention of HFC-23 is in the conclusion section. This seems out of place and, I'd suggest, inappropriate given that none of the information provided about HFC-23 is derived from data or analyses of observations presented in this manuscript and the points made aren't closely relevant for this manuscript.

This text has been moved to the introduction. We consider that it is not inappropriate to mention the association of HCFC-22 and HFC-23 with the linkage to the Clean

Development Mechanism and the relevant references.

(2) The discussion of HFCs being released predominantly in blends seems out of place and unusual. This is a straightforward conclusion based on uses of these gases by industry and it is not clear how the atmospheric data add to this discussion. There is a related point made in the conclusion about results not agreeing with some from Montzka et al (2014?), but there is no indication given as to the reason for this difference. Is it because the derived emissions disagree or is it because more information was brought to the analysis in the present manuscript than was available previously that defines our knowledge?

It is far from "a straightforward conclusion based on the use of these gases by industry". The only data from industry that are available are the compositions of the blends. All information about quantities, both of blends and individual HFCs, is commercially confidential. Supplementary information shows clearly the conclusion that global emissions calculated from atmospheric measurements are consistent with releases of HFCs wholly in blends. This is an important conclusion and not one that could have been obtained otherwise. It is not that we disagree with Montzka et al, 2015. In our analysis of the changing blends in the Supplementary material we were simply unable to confirm this hypothesis.

Details: Citation seems important but is lacking on line 50.

Reference (AFEAS, 2016).has been added.

Lines 61-65. Have no HFC results been reported by NOAA since 2004?

Montzka et al., 2015 reference has been added.

Precisions are quoted on lines 151-154 as single numbers, but I would guess that they have changed over time with different instruments and as atmospheric mole fractions have increased from v small levels. Does typical = median?

We have replaced lines 151-154 as follow:- "The GC-MS-Medusa measurement precisions for the four HCFCs and four HFCs are determined as the precisions of replicate measurements of the quaternary standards over twice the time interval as for sample-standard comparisons (Miller et al., 2008). Accordingly, they are upper-limit estimates of the precisions of the sample-standard comparisons. Typical daily precisions for each compound vary with abundance and individual instrument performance over time. Typical ranges for each compound measured between 2004 and 2016 are: for HCFC-22 (0.5 - 1.0 ppt); for -141b (0.05 - 0.1 ppt); for -142b; (0.05 - 0.1 ppt); for -124; (0.03 - 0.06 ppt); for HFC-134a (0.15 - 0.3 ppt), -125 (0.03 - 0.06 ppt), -143a (0.07 - 0.15 ppt) and -32 (0.04 - 0.2 ppt)."

Results and Discussion: How comparable are the model output mole fractions to the actual results? No indication of this is presented or mentioned.

We have included residual plots in the Supplementary Material which show the percentage difference between the model calculated mole fractions and the observed mole fractions.

Are growth rates quoted (line 237-238) based on some time interval, or just the measured change during 2015?

We have slightly modified this sentence but it does state in 2015. "The global mean mole fractions (pmol mol-1) observed in 2015, in descending order of abundance, are HFC-134a (83.3), HFC-125 (18.4), HFC-143a (17.7) and HFC-32 (10.5) with growth rates (pmol mol-1 yr-1 yr-1) for HFC-134a (5.6), HFC-125 (2.3), HFC-143a (1.5) and HFC-32 (1.6)."

Line 309, reconsider text. HCFC-141b growth rate isn't reported before 1998, so it doesn't seem accurate to suggest that emissions peaked in that year.

We apologies for this error. Our plots had been cut off at 1998, as this is the year in which the time series begin for several compounds. We have now extended these plots back to 1995, when in situ data began for some of these species. Figure 2 now shows

the first maximum in the growth rate of HFC-141b in 1998.

Consider units on increasing emission rates as per yr per yr.

Our Figures 3 and 4 refer just to emissions in Gg/yr.

Line 499 and 561-562. I believe this is correct only if you refer to relative rates of increase.

Text has been changed to reflect that rates are relative.

Line 565. "emissions of HCFC-22 represent 79% of the global cumulative HCFC burden: : :" doesn't make sense. Is the percentage relating to mole fractions or emissions? Please read text carefully it says "emissions"

WMO reports are appropriately cited by lead coauthor names; consider doing that as recommended in the reports.

WMO reports. We have altered the WMO references to reflect the Lead authors.